# MicroRNA-Based Discovery of Biomarkers, Therapeutic Targets, and Repositioning Drugs for Breast Cancer

**DOI:** 10.3390/cells12141917

**Published:** 2023-07-23

**Authors:** Qing Ye, Rebecca A. Raese, Dajie Luo, Juan Feng, Wenjun Xin, Chunlin Dong, Yong Qian, Nancy Lan Guo

**Affiliations:** 1West Virginia University Cancer Institute/Mary Babb Randolph Cancer Center, West Virginia University, Morgantown, WV 26506, USA; qiye@mix.wvu.edu (Q.Y.); rebecca.raese@gmail.com (R.A.R.); luodajie@hotmail.com (D.L.); june.wvu@gmail.com (J.F.); wenx@mix.wvu.edu (W.X.); lindadong2004@yahoo.com (C.D.); 2Health Effects Laboratory Division, National Institute for Occupational and Safety & Health, Morgantown, WV 26505, USA; yaq2@cdc.gov; 3Department of Occupational and Environmental Health Sciences, School of Public Health, West Virginia University, Morgantown, WV 26506, USA

**Keywords:** microRNA, breast cancer diagnosis, CRISPR-Cas9/RNAi, artificial intelligence, drug discovery, repositioning drugs, drug response

## Abstract

Breast cancer treatment can be improved with biomarkers for early detection and individualized therapy. A set of 86 microRNAs (miRNAs) were identified to separate breast cancer tumors from normal breast tissues (*n* = 52) with an overall accuracy of 90.4%. Six miRNAs had concordant expression in both tumors and breast cancer patient blood samples compared with the normal control samples. Twelve miRNAs showed concordant expression in tumors vs. normal breast tissues and patient survival (*n* = 1093), with seven as potential tumor suppressors and five as potential oncomiRs. From experimentally validated target genes of these 86 miRNAs, pan-sensitive and pan-resistant genes with concordant mRNA and protein expression associated with in-vitro drug response to 19 NCCN-recommended breast cancer drugs were selected. Combined with in-vitro proliferation assays using CRISPR-Cas9/RNAi and patient survival analysis, MEK inhibitors PD19830 and BRD-K12244279, pilocarpine, and tremorine were discovered as potential new drug options for treating breast cancer. Multi-omics biomarkers of response to the discovered drugs were identified using human breast cancer cell lines. This study presented an artificial intelligence pipeline of miRNA-based discovery of biomarkers, therapeutic targets, and repositioning drugs that can be applied to many cancer types.

## 1. Introduction

Breast cancer is the most common female cancer worldwide and accounts for 30% of new cancer cases for women each year in the United States. Despite the advances in breast cancer treatment, global breast cancer-related deaths are estimated to total 684,996 in 2020 [1]. The majority of fatalities from breast cancer are caused by metastatic disease. Some key unmet clinical needs for breast cancer treatment include: (1) Early detection: Since early identification enhances the likelihood of effective treatment, improved approaches for early detection of breast cancer are required, including more precise and accessible screening tools using minimally invasive liquid biopsies. (2) Precision medicine: It is essential to develop treatment strategies that are specifically tailored to the molecular features of each patient’s tumor. Although multiple breast cancer prognostic gene signatures [2,3,4] and molecular subtypes [5] have been applied in clinics, it is possible to enhance treatment outcomes and reduce unnecessary procedures by identifying biomarkers and creating targeted therapies utilizing current multi-omics techniques. (3) Metastatic and triple-negative breast cancer (TNBC): TNBC is the most aggressive breast cancer subtype with limited treatment options [6]. The 5-year survival rate of metastatic breast cancer remains low at 28%, in contrast to 86% to 99% for women with localized or regional breast cancer [7]. Refractory patients with failed prior therapies are generally put in hospice care. Developing more effective treatment strategies, including new and repositioning drugs, is important to prevent and treat metastatic breast cancer.

MicroRNAs (miRNAs) are a class of small non-coding RNAs that regulate gene expression and activate translation under certain circumstances. The dynamic interaction between miRNAs and their target genes is dependent on numerous factors, including miRNA subcellular location, target mRNA abundance, and miRNA-mRNA interaction affinity [8]. In addition to miRNA-target mRNA dynamics, alterations in miRNA-induced silencing complex (miRISC) localization in response to fluctuations in cellular environments, such as stress induced by heat shock and translation inhibition and serum starvation, can also affect miRNA activity and intracellular miRNA levels [9,10]. Extracellular miRNAs, enclosed and transported by exosomes, mediate cell-cell communication [11]. Both intracellular and extracellular miRNAs are pivotal in biogenesis, cellular signaling cascades, and numerous human diseases including cancer [12,13].

The investigation of miRNA expression and its implications in carcinogenesis has garnered significant attention within the field of cancer biology. For instance, several cancer-related pathways are regulated by miRNA, including apoptosis, differentiation, proliferation, and stem cell maintenance [14]. The utilization of miRNAs as biomarkers in cancer diagnostics has gained substantial attention, holding potential superiority over mRNAs in this context. First, the distinct miRNA expression profiles observed in different tissues and tumor types highlight the significant role they play in cancer diagnostics [15]. For example, Iorio et al. [16] identified a set of 15 miRNAs that were able to distinguish normal breast tissue from breast tumor samples with 100% accuracy. Researchers also noted differential miRNA expression based on various clinicopathologic features of the breast tumor samples such as estrogen and progesterone receptor status, positive lymph node metastasis, or higher proliferation indices. Additionally, researchers were able to classify the estrogen receptor status in 93 primary breast tumor samples [17]. Using this miRNA expression data and an independent test set of samples, this study also was able to classify basal-like and luminal A tumors according to their molecular subtype classification. Second, miRNAs can be isolated from a variety of samples in which mRNA extraction is usually difficult or unsuccessful such as formalin-fixed paraffin-embedded samples and serum or plasma samples [18,19,20]. A unique feature of miRNAs is their presence in biofluids, such as serum, plasma, saliva, and amniotic fluid [21]. A set of 26 miRNAs were identified in plasma samples that can separate normal and breast cancer patients [22]. Multiple breast cancer-derived exosomal miRNAs showed promise as liquid biopsy biomarkers to predict metastasis [23].

A single miRNA can regulate a broad spectrum of target protein-coding genes and subsequently direct entire cellular signaling pathways. Thus, modest changes in miRNAs can affect complex genetic, transcriptional, and translational networks, offering efficient therapeutic approaches in addition to diagnostic potential [24]. Despite the emerging opportunities, miRNA-based drugs need to overcome several technical difficulties, including the selection of appropriate delivery routes, management of in-vivo stability, targeting of specific tissues and cell types, and achievement of the desired intracellular effects [25]. Due to these practical challenges, there are currently no miRNA drugs for treating breast cancer. Efficient artificial intelligence (AI)/machine learning (ML) pipelines are needed to utilize miRNA/mRNA/protein expression data for effective screening of chemical compounds to identify new and repositioning drugs for breast cancer treatment.

The objective of this study is to identify relevant miRNAs shown to be differentially expressed in samples from breast cancer patients versus those from normal patients. By employing bioinformatics tools and statistical analyses, we sought to determine a select group of miRNA biomarkers that may be clinically useful in the diagnosis and prognosis of breast cancer. From the set of identified miRNA biomarkers, their experimentally validated target genes were selected for further analysis. Among these miRNAs and their target genes, pan-sensitive and pan-resistant biomarkers to 19 National Comprehensive Cancer Network (NCCN)-recommended drugs for treating breast cancer were identified. The proliferation potential of the target genes was assessed using public CRISPR-Cas9/RNAi screening data in Cancer Cell Line Encyclopedia (CCLE) human breast cancer cells. Potential new and repositioning drugs were discovered for treating breast cancer based on miRNA-regulated mRNA expression signature using Connectivity Map (CMap) [26,27]. Patient responders and non-responders to these potential new drug options were characterized by concordant mRNA/protein expression, non-silent mutations, and gene fusions in the genome-scale analysis of CCLE human breast cancer cell lines.

## 2. Materials and Methods

### 2.1. Patient Samples

The patient tissue samples consisted of snap-frozen breast tumors and normal breast tissue samples, which were stored at a temperature of −80 °C. Frozen patient blood samples collected in EDTA blood tubes were also stored at −80 °C until further use. All patient identifying information was removed from the pathology reports of the samples while retaining important details such as cancer stage, tumor grade, and histology. West Virginia University (WVU) Tissue Bank (Morgantown, WV), WVU Mary Babb Randolph Cancer Center (MBRCC) Biorepository, and the Cooperative Human Tissue Network (CHTN) were the providers of the de-identified patient tissue samples for this study. Comprehensive patient clinical information can be found in Table 1.

A published patient cohort from Iorio et al. [16] was used as an external validation set. This cohort contained 368 miRNA profiles quantified with microarray chips (KCI version 1.0) of seven normal breast tissue samples and 76 neoplastic breast tissue samples. The raw miRNA microarray data were available at EMBL’s European Bioinformatics Institute (EBI) with the accession number E-TABM-23.

### 2.2. RNA Isolation and Quality Assessment

From the frozen breast cancer tumors and normal breast tissue samples, total RNA was extracted using the mirVana miRNA Isolation kit following the manufacturer’s initial protocol. From the frozen blood samples, total RNA was isolated using a modified PAXgene protocol as described by Beekman et al. [28]. The thawed blood samples were transferred to PAXgene blood collection tubes, and total RNA extraction was performed using the PAXgene Blood miRNA kit according to the manufacturer’s instructions. The concentration of RNA was determined using the NanoDrop 1000 Spectrophotometer, while the quality of RNA was assessed using the 2100 Bioanalyzer. In the further analysis of this study, samples that met the quality control criteria were selected, including 52 tissue samples and 9 blood samples from the patients.

### 2.3. Microarray Analysis

The miRNA profiling with additional quality controls was conducted by Ocean Ridge Biosciences utilizing custom microarrays specifically designed with 1087 human miRNA probes. These miRNA arrays encompassed all 1098 human miRNAs documented in the Sanger Institute mirBASE version 15. The miRNA profiling analysis incorporated various quality control features, including negative controls, specificity controls, and spiking probes. To ensure the reliability of the data, a detection threshold was determined for each array by calculating the sum of five times the standard deviation of the background signal and the 10% trim mean of the negative control probes. Probes exhibiting consistently low signals were eliminated from the subsequent statistical analysis using these threshold values. The raw and processed miRNA profiles with the aforementioned normalization method were deposited to the NCBI Gene Expression Omnibus (GEO) with the accession number GSE37963.

### 2.4. The Cancer Genome Atlas (TCGA) Breast Cancer Patient Cohort

Prognostic analysis involving miRNA and mRNA data was performed on the breast cancer (BRCA) patient cohort obtained from The Cancer Genome Atlas (TCGA). The datasets were procured from the LinkedOmics database [29] (http://www.linkedomics.org/, accessed on 23 May 2023). This study used the normalized mRNA RSEM data comprising 1093 patient samples profiled with the Illumina HiSeq platform. Two sets of gene-level miRNA data were utilized: one derived from the Illumina Genome Analyzer (GA) platform (*n* = 324), and the other from the Illumina HiSeq platform (*n* = 755). Both miRNA datasets consisted of log_2_ normalized RPM values.

### 2.5. Proliferation Assays

Genes functionally involved in breast cancer cell proliferation were identified from CRISPR-Cas9 knockout and RNAi knockdown screening data. The whole-genome CRISPR-Cas9 screening data of human breast cancer cell lines (*n* = 48) were obtained from the DepMap 22Q4 data release [30,31,32] (https://depmap.org/portal, accessed on 23 May 2023). Additionally, RNAi screening data of human breast cancer cell lines (*n* = 34) were acquired from the project Achilles (https://depmap.org/R2-D2/, accessed on 23 May 2023). In this study, a normalized dependency score of less than –0.5 in either the CRISPR-Cas9 or RNAi screening was considered indicative of a significant knockout/knockdown effect.

### 2.6. Cancer Cell Line Encyclopedia (CCLE)

Genome-scale protein expression data of human breast cancer cell lines (*n* = 31) were obtained from the Gygi lab [33] (https://gygi.hms.harvard.edu/publications/ccle.html, accessed on 25 May 2023). These data were log2 transformed, and their mean expression levels were centered at 0. The mRNA sequencing data of breast cancer cell lines (*n* = 63) were acquired from the DepMap 22Q4 data release [30,31,32] (https://depmap.org/portal, accessed on 23 May 2023), and were also log2-transformed. Furthermore, gene mutations (*n* = 41,707) and fusion data (*n* = 2979) were retrieved from the DepMap 22Q4 data release.

### 2.7. Drug Sensitivity Data of Breast Cancer Cell Lines

The drug sensitivity data of breast cancer cell lines were obtained from two distinct sources. The first source was the Profiling Relative Inhibition Simultaneously in Mixtures (PRISM) [34] secondary screen data, which was provided in the DepMap 19Q4 data release (https://depmap.org/portal, accessed on 23 May 2023). The second source encompassed the Genomics of Drug Sensitivity in Cancer (GDSC1 and GDSC2) datasets [35,36,37], which were made available through CancerRxGene (https://www.cancerrxgene.org/, accessed on 25 May 2023). In this study, we examined drug sensitivity using various measurements, including IC_50_, ln(IC_50_), EC_50_, and ln(EC_50_). Further details regarding the categorization of drug sensitivity and resistance can be found in our previous work [38,39].

### 2.8. Bioinformatic Tools

TarBase database v7.0 [40] (https://dianalab.e-ce.uth.gr/html/universe/index.php?r=tarbase, accessed on 23 May 2023) and v8.0 [41] (https://dianalab.e-ce.uth.gr/html/diana/web/index.php?r=tarbasev8, accessed on 23 May 2023) was used to find experimentally validated microRNA-target interactions of our selected miRNAs.

ToppGene [42] (https://toppgene.cchmc.org/, accessed on 6 July 2023) is a comprehensive bioinformatics tool designed to facilitate gene list analysis and functional interpretation. It employs a variety of data mining and statistical techniques to uncover significant biological associations and gain insights into the functional relevance of gene sets. ToppGene provides a user-friendly platform for researchers to prioritize genes, unravel biological pathways, identify key biological functions, and discover potential disease associations. In this study, ToppGene was utilized to detect the functional enrichment of selected gene lists.

For drug repositioning analysis, we employed the bioinformatic tool Connectivity Map (CMap, https://clue.io/cmap, accessed on 23 May 2023) [26,27]. CMap utilizes gene expression profiles to determine connectivity scores based on drug-induced transcriptional profiles. The connectivity scores range from –1 to 1, with 1 indicating the highest degree of expression similarity. In this study, we considered a raw connectivity score higher than 0.9 with a *p*-value < 0.05 as a significant result.

Cytoscape version 3.9.1 [43] (https://cytoscape.org/, accessed on 15 June 2023) was utilized to visualize the network results of miRNAs and their target genes.

### 2.9. Statistical Analysis

The Significance Analysis of Microarrays (SAM) method was used to identify miRNA markers displaying differential expression patterns in breast cancer samples. The following criteria were used to select statistically significant differentially expressed miRNAs: (1) an expression change threshold of >2 or <0.5 between normal and tumor tissues, or between blood samples from normal and cancer patients; (2) *p*-values less than 0.05 (unpaired or paired *t*-tests depended on the specific sample set being analyzed); and (3) a false discovery rate (FDR) below 0.05. Comparisons between two groups were assessed using two-sample *t*-tests. Survival analysis was conducted utilizing the Kaplan-Meier method implemented in the “survival” package (version 3.5.3). In the survival analysis, the log-rank test *p*-value was used to evaluate the difference in survival probabilities. Prognosis analysis was carried out employing the univariate Cox model. RStudio (version 2023.03.1 Build 446) with R version 4.2.1 was the primary statistical analysis tool throughout this study.

## 3. Results

### 3.1. Differential MiRNA Expression in Breast Cancer Tissue and Blood

Using our patient cohort, we identified a collection of 86 miRNAs exhibiting significant differential expression when comparing tumor and normal breast tissue samples. To provide a visual representation of the expression patterns, Figure 1 showcased a heatmap illustrating the clustering results of patient samples based on the expression profiles of these 86 miRNAs. In Figure 1, four breast cancer samples were mistakenly classified into the normal breast tissue group, and one normal breast tissue sample was misclassified into the breast cancer group. The overall accuracy in the unsupervised classification was 90.4% using the expression profiles of 86 miRNAs. Here the denoted breast cancer in Figure 1 referred to the cases with histology of ductal adenocarcinoma, invasive ductal carcinoma, invasive lobular carcinoma, metaplastic carcinoma, mucinous carcinoma, and predominantly lobular listed in Table 1. The detailed fold changes of these 86 miRNAs in patient tumor tissues were provided in Appendix A.

Subsequently, we investigated the prognostic significance of the identified 86 markers. Notably, 12 specific miRNAs shown in Table 2 displayed consistent diagnostic and prognostic relevance within both our patient cohort and TCGA breast cancer patients. In this analysis, miRNAs sharing the same prefix were analyzed together, i.e., the results were not miRNA isoform specific. Our findings revealed that five miRNAs exhibited potential oncogenic characteristics (i.e., over-expression in tumors and survival hazard), while seven miRNAs demonstrated potential tumor suppressor properties in breast cancer (i.e., under-expression in tumors and survival protective).

Out of the initial set of 86 markers, a total of 33 miRNAs exhibited statistically significant differential expression between matched tumors and normal breast tissue samples from the same patients. The clustering analysis of matched samples demonstrated a distinct separation with an overall classification accuracy of 100%, wherein all tumor tissues and all normal tissues were observed to cluster in different groups, as depicted in Figure 2. This clustering pattern indicated a clear delineation between the gene expression profiles of breast cancer and normal samples. The fold changes of these 33 miRNAs in patient blood samples were provided in Appendix A.

Among the analyzed miRNA markers, a subset of six markers exhibited concordant differential expression between breast cancer vs. normal in both tissues and blood samples. Utilizing these six markers, the breast cancer and normal blood samples could be effectively distinguished with an overall classification accuracy of 100%, as illustrated by the heatmap depicted in Figure 3. This finding further underscored the potential of these markers in accurately discriminating between breast cancer and normal patient samples using minimally invasive blood tests. Detailed fold change information in blood samples was provided in Appendix A.

Next, we examined the miRNAs that displayed consistent differential expression patterns in our samples as well as in the cohort from Iorio et al. [16]. A total of four miRNAs (up-regulated miR-155, and down-regulated miR-204, miR-145, and miR-143) demonstrated statistically significant differential expression between tumors vs. normal breast tissues (FDR < 0.05 in SAM, *p* < 0.05; unpaired *t*-tests) in both the Iorio dataset (*n* = 83) and our cohort (*n* = 52). MiRNAs showing concordant expression patterns in breast cancer initiation and progression among our cohort, Iorio et al. [16], and TCGA were depicted in Figure 4. MiR-204 was a potential tumor suppressor based on the results from all three patient cohorts. These findings emphasized the robustness and reproducibility of the observed miRNA expression changes across different datasets and further supported the potential relevance of these miRNAs in breast cancer pathogenesis. The detailed fold change information was provided in Appendix A.

### 3.2. Association with Drug Sensitivity

Among the 86 miRNAs identified in this study, a subset of 23 miRNAs was found in TarBase (v7 or v8) with experimentally validated target genes. These miRNAs collectively targeted a total of 4117 genes, with published experimental evidence that was curated and stored in TarBase (Appendix A). To gain further insights into the clinical relevance of these miRNAs and their respective target genes, we investigated their associations with drug sensitivity in breast cancer treatment.

For drug sensitivity analysis, we classified the breast cancer cell lines as sensitive or resistant to each drug based on the measurements of IC_50_, ln(IC_50_), EC_50_, and ln(EC_50_), following the classification method described in our previous publications [38,39]. Both mRNA and protein expression levels were considered in determining the association with drug sensitivity. A gene was deemed sensitive to a drug if it exhibited significantly higher expression (*p* < 0.05; two-sample *t*-tests) in the sensitive cell line group. Conversely, a gene was classified as resistant to a drug if it demonstrated significantly higher expression (*p* < 0.05; two-sample *t*-tests) in the resistant cell line group.

Additionally, we defined pan-sensitive genes as those exhibiting sensitivity or non-resistance to all 19 breast cancer drugs selected according to the NCCN guidelines (shown in Table 3), in the PRISM, GDSC1, and GDSC2 data. Conversely, pan-resistant genes were defined as those displaying resistance or non-sensitivity to all 19 breast cancer drugs in the aforementioned datasets. Table 3 presented the pan-sensitive and pan-resistant genes at both mRNA and protein levels, along with the pan-sensitive and pan-resistant miRNAs.

Among all the miRNAs and genes included in Table 3, we depicted miRNA-target genes and their association with pan-sensitivity/pan-resistance to 19 NCCN-recommended breast cancer drugs in Figure 5. Furthermore, significantly enriched cytobands and gene families for pan-sensitive and pan-resistant genes were analyzed with ToppGene (Figure 6). There were no significantly represented gene families for pan-sensitive genes. No significant pathways were enriched for either pan-sensitive or pan-resistant genes (Table 3). Through these comprehensive analyses, we aimed to uncover potential associations between the identified miRNAs, their target genes, and drug sensitivity, providing valuable insights into their roles in breast cancer biology and therapeutic response.

### 3.3. Discovery of Potential Therapeutic Targets and Repositioning Drugs

Since no significantly enriched pathways were found for pan-sensitive or pan-resistant genes (Table 3), we further included patient survival genes and in-vitro proliferation genes to refine the gene list. To identify therapeutic targets and candidate compounds for treating breast cancer, we designed the mechanism of drug action to maintain the expression of non-proliferative survival-protective/pan-sensitive genes and suppress the survival-hazard/pan-resistant genes (Figure 7A). Among the 4117 genes targeted by our diagnostic miRNAs, we performed a refined selection to identify 28 up-regulated and 28 down-regulated gene lists for input into the CMap analysis. The 28 up-regulated genes met the following criteria: (1) exhibited a significant hazard ratio < 1 (log-rank *p* < 0.05) in survival analysis of TCGA-BRCA; (2) were classified as pan-sensitive genes for all 19 NCCN breast cancer drugs at both the mRNA and protein levels in CCLE BRCA cell lines; (3) did not display significant dependency scores (<–0.5) in more than 50% of the BRCA cell lines in both CRISPR-Cas9 and RNAi screening data. The 28 down-regulated genes met the following criteria: (1) demonstrated a significant hazard ratio > 1 (log-rank *p* < 0.05) in survival analysis of TCGA-BRCA; (2) were classified as pan-resistant genes for all 19 NCCN breast cancer drugs at both the mRNA and protein levels in CCLE BRCA cell lines. Detailed information was provided in Appendix A.

We input the selected 28 up-regulated genes and the 28 down-regulated genes into CMap. From the CMap output, we selected significant (raw connectivity score > 0.9, *p* < 0.05) pathways and compound sets as heuristic therapeutic targets. We further examined the half-maximal inhibitory concentration (IC_50_) and half-maximal effective concentration (EC_50_) of the significant compounds in the PRISM dataset, which led to the identification of 20 potential new or repurposed drugs with available IC_50_/EC_50_ measurements in the PRISM dataset. Our pipeline for repositioning drug discovery was shown in Figure 7A. Figure 7B displayed the boxplots of IC_50_/EC_50_ values of our discovered drugs in BRCA cell lines. Some of the drugs contained some outliers with extremely large values. The outliers were removed from the analysis and were not shown in the plot. These drugs belonged to 11 compound sets. In addition, three pathways were identified in shRNA knockdown experiments in breast adenocarcinoma MCF7 cells that matched our CMap input 56-gene expression signature (connectivity score > 0.9, *p* < 0.05), including knockdown of heat shock 70 (Hsp70) proteins (KD_HEAT_SHOCK_70KDA_PROTEINS:MCF7:TRT_SH.CGS) including *HYOU1*, *HSPA5*, *HSPA9*, and *HSPA14* genes, SOD pathway (BIOCARTA_SODD_PATHWAY:MCF7:TRT_SH.CGS) including *BAG4, BIRC3, CASP8, FADD, RIPK1, TNF, TNFRSF1A, TNFRSF1B, TRADD*, and *TRAF2*, and knockdown of A-Kinase-Anchoring-Proteins (AKAPs; KD_A_KINASE_ANCHORING_PROTEINS:MCF7:TRT_SH.CGS) including *AKAP9, AKAP13*, and *AKAP11*. Details were provided in Appendix A.

Furthermore, we explored the genome-scale resistant and sensitive genes associated with our discovered drugs. Table 4 presented the selected sensitive and resistant genes for each drug, which had concordant expression at both mRNA and protein levels in the studied human breast cancer cell lines. Additionally, we identified gene fusions and mutations associated with the categorized sensitive and resistant cell lines to the corresponding drugs, highlighting the top fusion/mutation genes in each cell line (Appendix A). By integrating CMap analysis with our comprehensive multi-omics and drug sensitivity investigation, we unveiled potential novel therapeutic options and characterized patient responders/non-responders for these new breast cancer drugs (Figure 7A).

MiR-204 was identified as a potential tumor suppressor as under-expressed in tumors in our patient cohort and samples from Iorio et al. [16] as well as patients with a poor prognosis in TCGA-BRCA (Table 2 and Figure 4). MiR-204 was associated with drug sensitivity to carboplatin and tucatinib, and drug resistance to docetaxel, ixabepilone, paclitaxel, and vinorelbine. In addition, miR-204 was associated with resistance to RITA, a potential new drug option for treating breast cancer (Appendix A). MiR-204-3p and miR-204-5p targeted genes were provided in Appendix A. Among these target genes, those that were pan-sensitive or pan-resistant to 19 NCCN-recommended breast cancer drugs (Table 3) were shown in Figure 8. MiR-204 targeted genes (*ACSL4* and *WLS*, Table 4) that were sensitive to BRD-K12244279 discovered for treating breast cancer in this study were also shown in Figure 8.

## 4. Discussion

This study identified 86 miRNAs differentially expressed between tumors and normal breast tissue samples in molecular classification. Four miRNA isoforms were confirmed in an external patient cohort as potential tissue-based breast cancer diagnostic biomarkers. Six miRNAs had concordant differential expression between breast cancer and normal samples in both tissue and blood, with miR-30a*, miR-224*, miR-154 downregulated and miR-155, miR-1972, miR-3172 upregulated in breast cancer. Twelve miRNAs had concordant expression patterns in our collected tumors vs. normal breast tissues and TCGA-BRCA patient survival hazard ratios, with five as potential oncogenes and seven as potential tumor suppressors. These results warranted further independent validation to substantiate the clinical utility of the identified miRNAs for breast cancer diagnosis using biopsies and/or liquid biopsies.

Among seven potential tumor-suppressing miRNAs identified in this study, high expression of miR-100 was associated with better outcomes in women with luminal A tumors treated with adjuvant endocrine therapy and was inversely linked to mRNA expression of *PLK1*, *FOXA1*, *mTOR*, and *IGF1R* [44]. By targeting *FOXA1*, miR-100 suppressed the migration, invasion, and proliferation of breast cancer cells [45]. MiR-101-5p, a tumor suppressor, triggered apoptosis in HER2+ breast cancer and sensitized initially resistant cells to lapatinib and trastuzumab [45,46]. MiR-101 targeted Janus kinase 2 (JAK2) in inhibiting proliferation and promoting apoptosis of breast cancer cells [47]. MiR-204 was significantly under-expressed in breast cancer tumors in our patient cohort and samples from Iorio et al. [16], as well as breast cancer patients with a poor survival outcome in TCGA-BRCA (Figure 4). MiR-204-5p-regulated *PI3K/Akt* signaling inhibited tumor growth, metastasis, and immune cell reprogramming in breast cancer [48]. By inhibiting miR-204-5p and enhancing *RRM2* expression, *DSCAM-AS1* promoted proliferation and suppressed apoptosis of breast cancer cells, indicating the therapeutic potential of *DSCAM-AS1*/miR-204-5p/*RRM2* [49]. MiR-204-5p targets pan-sensitive genes (*HELLS* and BAZ1B), and multiple pan-resistant genes, including MAVS, GSK3A, and *PEG10* identified in this study. MiR-204-3p targets a pan-resistant gene ZNF24 (Figure 8). MiR-205 is a recognized tumor suppressor in breast cancer, with decreasing expression in breast cancer initiation, progression, metastasis, increased stemness [50], and more aggressive molecular subtypes [51]. MiR-205 was also identified as pan-sensitive to 19 NCCN-recommended breast cancer drugs in this study. MiR-30 family attenuated breast cancer invasion and bone metastasis as a promising therapeutic target for triple-negative breast cancer [52]. In this study, miR-30a* was under-expressed in both breast cancer tumors and blood samples compared with the corresponding normal controls, indicating its potential as a diagnostic biomarker. MiR-379 potently suppressed breast cancer formation and growth, partly through the regulation of COX2 [53] and inhibition of proliferation [54]. MiR-379 and miR-204, both identified in this study as potential oncosuppressors, are key regulators of TGF-β-induced IL-11 production, important in breast cancer bone metastasis [55]. MiR-99a inhibited breast cancer tumorigenesis and progression by targeting the mTOR/p-4E-BP1/p-S6K1 pathway [56] and the cell-cycle pathway through downregulating CDC25A [57]. MiR-99a-5p sensitized breast cancer cells to doxorubicin by regulating the COX-2/ABCG2 axis [58]. Our identified seven miRNAs are supported by the literature as BRCA tumor suppressors.

For the five identified potential oncomiRs, miR-193a-3p promoted tumor progression by targeting GRB7, ERK1/2, and FOXM1 signaling pathways [59] and *PTP1B* [60] in HER2-positive breast cancer cells. MiR-301b promoted cell proliferation by regulating *PRKD3* in ER-mutant breast cancer, accounting for up to 30% of metastatic ER-positive breast cancer [61]. MiR-301b exerted tumor-promoting effects through co-regulation with its target gene *NR3C2* in breast cancer MCF7 and BCAP-37 cells [62]. A miRNA in the same family, miR-301a, was identified as pan-resistant to 19 NCCN-recommended breast cancer drugs in this study. MiR-615-3p contributed to epithelial-to-mesenchymal transition and metastasis in breast cancer by regulating a negative feedback loop involving the PICK1/TGFBRI axis [63]. Despite the association between miR-7 expression and patient poor survival, miR-7 inhibited breast cancer spreading and tumor-associated angiogenesis in metastatic breast cancer [64]. By modifying *KLF4*, miR-7 prevented breast cancer stem-like cells from metastasizing to the brain [65]. Via activating the ERK signaling pathway, ADAM8 induced miR-720 expression, which in turn promoted the aggressive phenotype of triple-negative breast cancer cells [66]. Overall, the literature supports our identified breast cancer oncomiRs, except for miR-7.

Upon the validation of our identified 86 miRNAs in multiple public patient cohorts, their experimentally validated target genes were retrieved with TarBase. Further analysis of these target genes pinpointed pan-sensitive and pan-resistant genes to 19 NCCN-recommended drugs for treating breast cancer. In the integrative analysis of the public in-vitro proliferation screening assay data of CCLE and TCGA-BRCA patient survival, a 56-gene expression signature was constructed to discover new and repositioning drugs for improving breast cancer treatment and survival outcomes using CMap. Through the CMap analysis, compounds that can inhibit the input down-regulated genes and maintain the expression of the input up-regulated genes in breast cancer cells were identified. The significant pathways and compound sets (raw connectivity score > 0.9, *p* < 0.05) were considered valid hypotheses worthy of further investigation. From these significant results, we further examined the compounds with average IC_50_/EC_50_ measurements less than 7 µM in human breast cancer cell lines as potential drugs for breast cancer treatment and were discussed as follows.

PI3K inhibitor SAR245409 combined with MEK inhibitor AS703026 (pimasertib) synergistically magnified anti-proliferative effects in TNBC [67]. Lestaurtinib, a tyrosine kinase inhibitor, enhanced the in-vitro drug effects of the PARP1 inhibitor AG14361 in breast cancer treatment, partly by suppressing NF-κB signaling [68]. Midostaurin is an FDA-approved multi-targeted protein kinase inhibitor for the treatment of both solid and non-solid tumors [69]. Midostaurin preferentially suppressed the proliferation of TNBC cells among breast cancer cell lines by inhibiting the Aurora kinase family [70]. Nilotinib (Tasigna) is an approved chronic myelogenous leukemia drug. Nilotinib can reduce doxorubicin-induced cardiac impairment [71]. Breast cancer cells resistant to tamoxifen therapy were resensitized by a combination of sorafenib and nilotinib via the estrogen receptor [72]. MEK inhibitor CI-1040 (PD184352) exhibited notable antitumor activity in preclinical models, specifically against pancreatic, colon, and breast cancers, and was well tolerated in Phase I clinical trials [73] but had insufficient efficacy in Phase II clinical studies [74]. A second-generation MEK inhibitor PD0325901 was also selected using machine learning methods as a drug for breast cancer treatment [75] and amplified anti-proliferative and anti-clonogenic effects of gefitinib and AT7867 by activating apoptosis in TNBC cells [76]. PD0325901 has much improved pharmacologic and pharmaceutical properties compared with CI-1040 and has also entered clinical development [74]. Combination therapy of MEK inhibitor PD98059 and anti-diabetic drug Rosiglitazone caused invasive and spreading cancer cells to transform into post-mitotic adipocytes, which inhibited the invasion of the primary tumor and the development of metastases [77]. U0126 is a highly potent and selective inhibitor specifically targeting MAPK, MEK1, and MEK2 signaling pathways and plays a pivotal role in maintaining cellular homeostasis [78]. U0126 reduced hyperpolarized pyruvate to lactate conversion, a non-imaging method for detecting tumors and treatment response, in breast cancer cells [79]. U0126 can reduce breast cancer cell content in the S phase, suggesting anti-proliferative effects by blocking the cell cycle [80]. ERK phosphorylation in T47D breast cancer cells exhibited resistance to MEK inhibition by U0126, PD98059, and PD198306 [81]. PD198306 has not been reported as a breast cancer drug that can prolong patient survival outcomes.

RITA small-molecule anticancer drug specifically targeting p53 [82] has been extensively studied in breast cancer [83]. MEK inhibitor Selumetinib demonstrates clinical benefits in treating pediatric neurofibromatosis type I in phase II clinical trials [84]. Selumetinib inhibits cell proliferation/migration, induces apoptosis/G1 arrest [85], and prevents lung metastasis [86] in TNBC. Serdemetan (JNJ-26854165) is a small-molecule antagonist of MDM2, exhibiting antiproliferative activity in a range of tumor cell lines characterized by wild-type *p53* [87]. In a Phase I clinical study, Serdemetan was well tolerated by patients with advanced solid tumors with exposure-related QTc liability, and partial response was observed in a breast cancer patient [88]. Sunitinib is an FDA-approved oral inhibitor of multi-target receptor tyrosine kinases and has demonstrated efficacy in the treatment of renal cell carcinoma and imatinib-resistant gastrointestinal stromal tumor [69]. Sunitinib achieved an overall response rate of 11% in treating metastatic breast cancer in a Phase II clinical trial [89]. Y-27632, a rho-kinase inhibitor, attenuates breast cancer cell migration, proliferation, and bone metastasis [90]. MEK inhibitors BRD-K12244279 and PD98059 were identified as repurposing drug agents for treating vestibular schwannoma [91]. Pilocarpine is a cholinergic parasympathetic stimulant, generally used to treat xerostomia and oral cavities [92]. Tremorine was utilized to induce tremors and replicate symptoms resembling Parkinson’s disease in animal models [93]. Overall, 16 protein kinase inhibitors selected by our AI pipeline are promising targeted therapies for treating breast cancer, substantiating the effectiveness of our AI/ML methods. We discovered experimental agents, including MEK inhibitors PD19830 and BRD-K12244279, pilocarpine, and tremorine as potential new drug options for improving breast cancer survival outcomes, which were previously unknown.

Characterization of patient responders/non-responders is essential in designing clinical trials to test the efficacy of new and repositioning drugs. HCC1143 is a top in-vitro model to investigate TNBC and its transcriptional profiles are suitable to study the Interferon, *IGF1*, and *MET* signaling pathways [94]. HCC1143 was sensitive to PD0325901, PD184352, PD198306, pilocarpine, RITA, serdemetan, tremorine, and Y-27632, and resistant to lestaurtinib, midostaurin, and nilotinib. HCC1806, another TNBC cell line with multi-drug resistance [95], was sensitive to midostaurin, nilotinib, PD0325901, PD198306, PD98059, pilocarpine, RITA, sorafenib, TG101348, tremorine, and Y-27632, and was resistant to AS703026, dasatinib, and lestaurtinib. HCC1937, a TNBC line derived from a 24-year-old woman with a family history of breast cancer and a germline mutation in *BRCA1* [96,97], was sensitive to pilocarpine, RITA, TG101348, and tremorine, and resistant to AS703026, dasatinib, midostaurin, nilotinib, PD0325901, PD198306, and PD98059. TNBC HCC38 cell line was sensitive to PD98059, pilocarpine, RITA, serdemetan, sorafenib, tremorine, and Y-27632, and resistant to dasatinib, lestaurtinib, midostaurin, nilotinib, PD0325901, PD184352, and PD198306. Luminal A MCF7 cells were sensitive to PD184352, PD198306, PD98059, RITA, TG101348, and U0126, and resistant to dasatinib, BRD-K12244279, and PD0325901. These results are useful for the clinical development of these drugs for treating breast cancer, especially TNBC.

Top occurring gene fusions and non-silent mutations in the above cell lines were pinpointed. The top fusion genes in HCC1143 were *PPP2R5A* and *ZNRD1ASP*, both having 8 fusions, and the top mutated gene is *MUC3A* with 6 mutations. *SMURF2* had 6 fusions and *TTN* had 10 non-silent mutations in HCC1806. *MUC3A* had 7 mutations in HCC1937. *COL24A1* had 12 fusions, and *MUC3A* and *MUC5AC* both had 6 mutations in HCC38. In MCF7, the top fusion genes were *ATXN7* and *VMP1*, both having 6 fusions, and the top mutated gene was *MUC3A* with 7 mutations. *MUC3A* was the top mutated gene in multiple BRCA cell lines and promoted the progression of colorectal cancer through the PI3K/Akt/mTOR pathway [98]. *RAD51C-ATXN7* fusion gene expression was associated with functional damage of DNA repair and its fusion transcript generated a fusion protein in colorectal tumors [99]. *RAD51C-ATXN7* fusion was also present in MCF7. More details were provided in Appendix A. The multimodal information presented in this study is important for future research and clinical management of breast cancer. The AI pipeline presented in this study can also be applied to model other data types for new/repositioning drug discovery, such as DNA copy number variation, transcriptomic, and proteomics profiles in bulk tumors and single cells for other cancer types as we previously published [100,101,102]. In the future, we will also apply this pipeline to model other structural variants for the discovery of biomarkers and therapeutic targets.

This study has several limitations. First, our patient cohort has a small sample size. The identified tissue-based 86 miRNAs and six blood-based miRNAs can separate breast cancer and normal samples with an overall accuracy of 90.4% and 100%, respectively. Nevertheless, due to the small sample size, their clinical utility in breast cancer diagnosis needs to be substantiated with the following studies: (1) an independent external patient cohort with a larger sample size; (2) high overall accuracy, specificity, and sensitivity in the external validation with the classification model fixed in the training-validation process; and (3) prospective validation of clinical benefits showing improved patient outcomes by using the diagnostic miRNA tests. Fulfilling these tasks will be time-consuming and require a vast amount of resources. Second, the functions of the identified potential oncomiRs and potential tumor-suppressing miRNAs need to be investigated in future research. Following the mechanistic characterization, the development of miRNA-based drugs needs to overcome technical challenges in achieving the optimal delivery route, intracellular efficacy, tissue specificity, and in-vivo stability. Finally, the identified compounds as new or repositioning drugs for breast cancer treatment were based on bioinformatics analysis and in-vitro IC_50_/EC_50_ measurements in human breast cancer cell lines. Their efficacy in treating breast cancer needs to be tested in future animal studies and/or clinical trials before obtaining regulatory approvals for clinical applications.

## 5. Conclusions

This study identified a set of 86 miRNAs to distinguish breast cancer tumors from normal breast tissues. Six miRNAs had concordant expression in both tumors and breast cancer patient blood samples compared with the normal control samples, with implications for the development of minimally-invasive diagnostic tests using liquid biopsies. Twelve miRNAs had concordant expression patterns in breast cancer initiation and progression, with seven as potential tumor suppressors and five as potential oncomiRs. From experimentally validated target genes of these 86 miRNAs, pan-sensitive and pan-resistant genes with concordant mRNA and protein expression associated with in-vitro drug response to 19 NCCN-recommended breast cancer drugs were selected. These genes, combined with in-vitro proliferation assays and patient survival analysis, led to the discovery of MEK inhibitors PD19830 and BRD-K12244279, pilocarpine, and tremorine as potential new drug options for treating breast cancer. Multi-omics biomarkers for response to the discovered drugs were identified using CCLE breast cancer cell lines. The presented AI pipeline utilizing multi-omics data analysis for the discovery of biomarkers and drug development can be applied to many human cancers.

## 6. Patents

The results in this manuscript were included in a provisional patent with the serial number US 63/515,087. 

## Figures and Tables

**Figure 1 cells-12-01917-f001:**
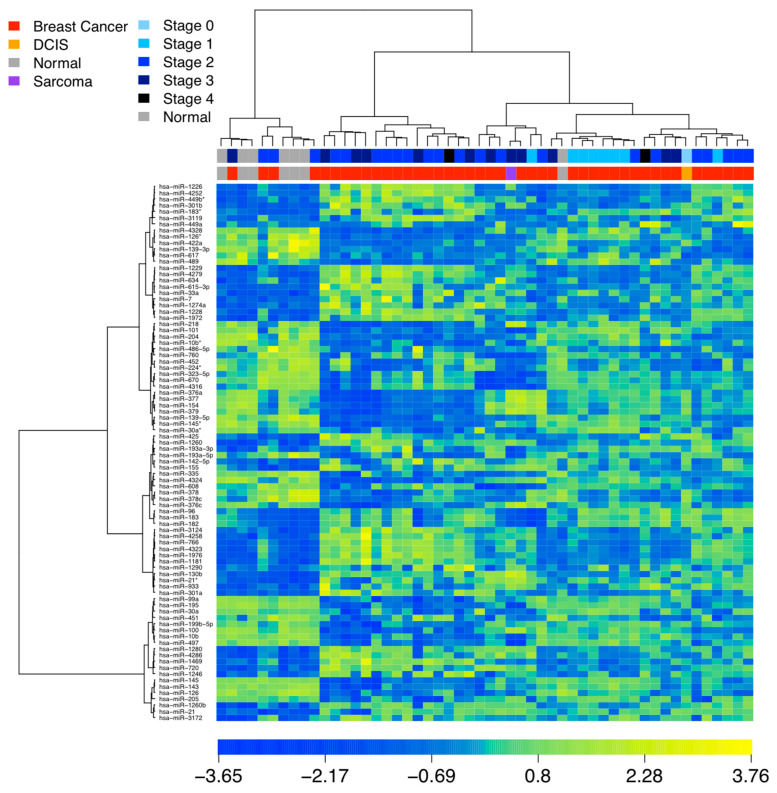
Heat map of miRNA-based clustering of patient samples (*n* = 52). 86 miRNA markers exhibiting significant differential expression in tissue (FDR < 0.05 in SAM, fold change > 2 or <0.5; unpaired or paired *t*-tests).

**Figure 2 cells-12-01917-f002:**
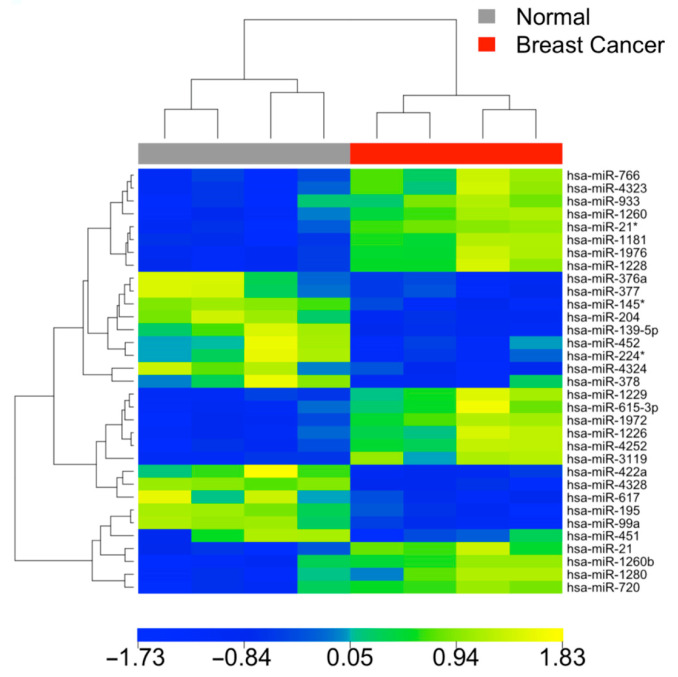
Heatmap of matched tumors and normal tissue samples from the same four breast cancer patients, showing separation of expression profiles in tumors and normal tissues (FDR < 0.05 in SAM, fold change > 2 or <0.5; in paired *t*-tests).

**Figure 3 cells-12-01917-f003:**
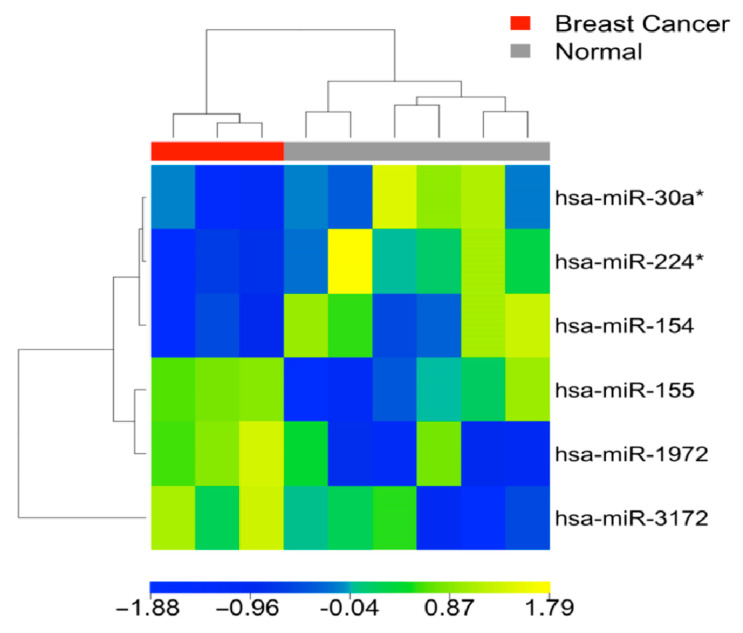
Heatmap of blood samples showing clustering analysis of six miRNAs concordantly expressed between breast cancer and normal samples in tissues and blood (FDR < 0.05 in SAM, *p* < 0.05 in unpaired *t*-tests). Blood samples from breast cancer patients (*n* = 3) can distinctly separate from those from normal individuals (*n* = 6).

**Figure 4 cells-12-01917-f004:**
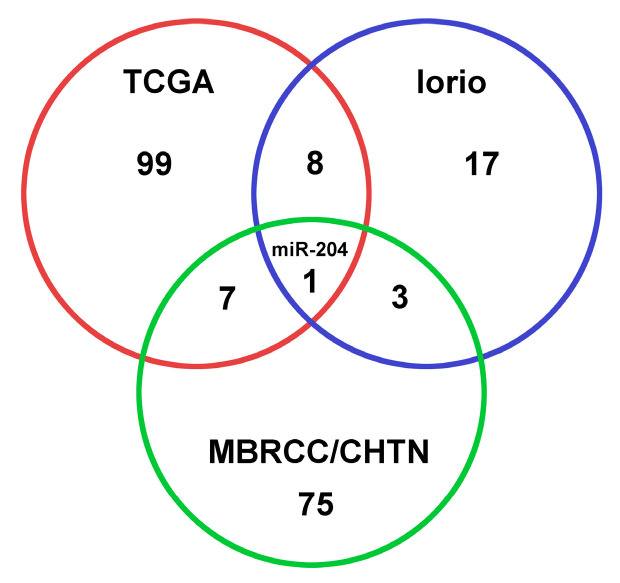
Venn diagram of selected miRNAs (isoform-specific) across different patient cohorts. The miRNA selection in the Iorio cohort and the MBRCC/CHTN breast cancer cohort was based on the statistically significant differential expression observed between breast cancer tumors and normal breast tissue samples. In TCGA, miRNA expression association with breast cancer progression was indicated with a hazard ratio (HR). HR > 1: survival hazard miRNA; HR < 1: survival protective miRNA. Detailed information was provided in Appendix A.

**Figure 5 cells-12-01917-f005:**
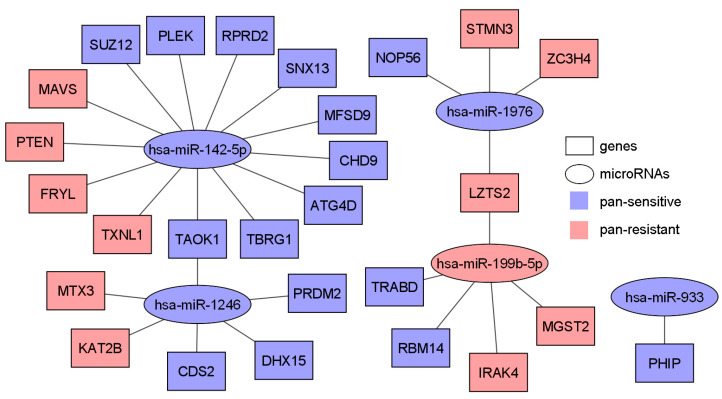
MiRNA-target genes and their association with pan-sensitivity/pan-resistance to 19 NCCN-recommended breast cancer drugs. The miRNAs and their experimentally validated target genes were included in Table 3.

**Figure 6 cells-12-01917-f006:**
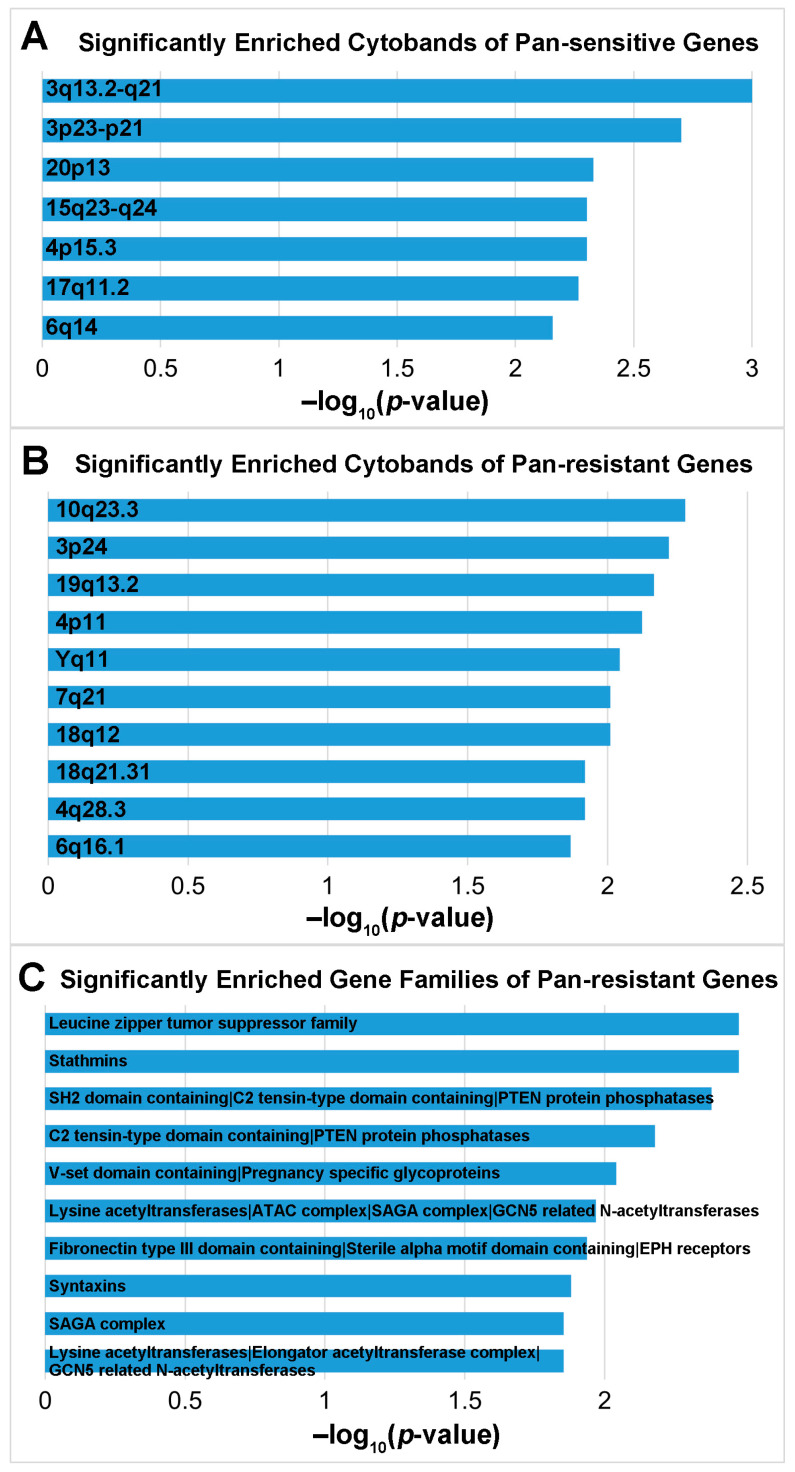
Significantly enriched cytobands and gene families of pan-sensitive and pan-resistant genes. (**A**) The –log_10_(*p*-value) of the significantly enriched cytobands of the pan-sensitive genes. (**B**) The –log_10_(*p*-value) of the top 10 significantly enriched cytobands of the pan-resistant genes. (**C**) The –log_10_(*p*-value) of the top 10 significantly enriched gene families of the pan-resistant genes. Details were provided in Appendix A.

**Figure 7 cells-12-01917-f007:**
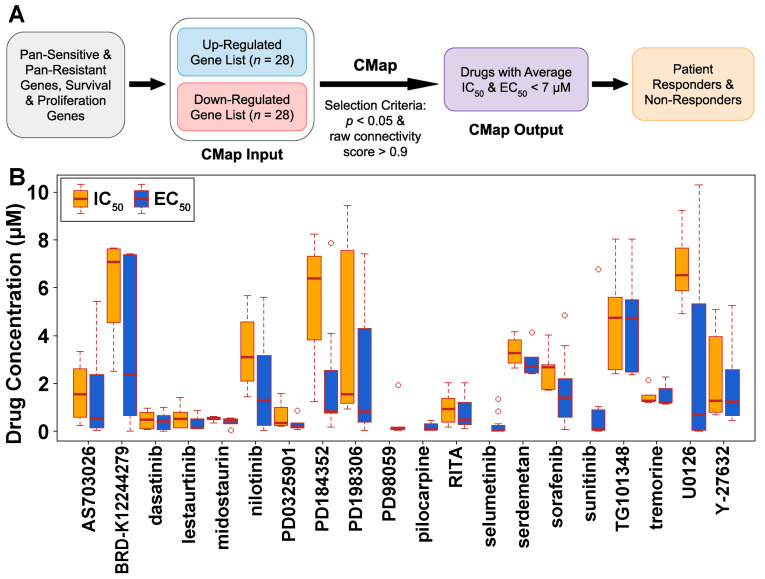
Discovery of potential new and repositioning drugs for treating breast cancer. (**A**) The flowchart of our AI pipeline for drug discovery. (**B**) The distribution of drug concentrations (IC_50_/EC_50_) of the 20 selected drugs in the PRISM data. All selected compounds had average IC_50_ and EC_50_ < 7 µM in the tested human breast cancer cell lines after excluding outliers.

**Figure 8 cells-12-01917-f008:**
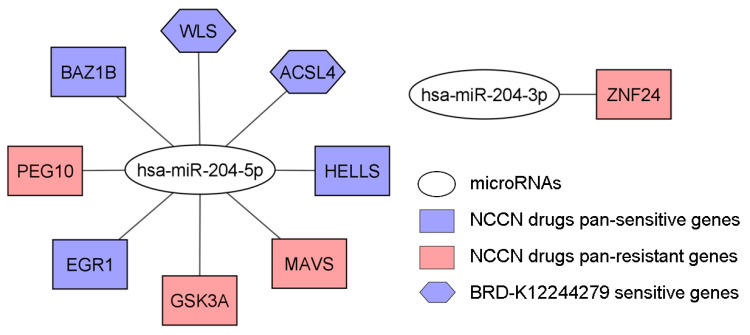
Hsa-miR-204 targeted genes. Among the hsa-miR-204-3p and hsa-miR-204-5p targeted genes, those pan-sensitive/pan-resistant to 19 NCCN breast cancer drugs and sensitive to BRD-K12244279 were shown.

**Table 1 cells-12-01917-t001:** Clinical information of patient samples used in this study.

Clinical Information	Sample Size (*N*)
Cancer Stage	
Ductal carcinoma in situ (DCIS)	1
1	8
2	22
3	12
4	2
Normal	7
Age	
<60	27
60–80	17
>80	3
Missing	5
Tumor Grade	
1	3
2	13
3	25
Missing	11
Histology	
Ductal adenocarcinoma	1
In situ carcinoma	1
Invasive ductal carcinoma	35
Invasive lobular carcinoma	3
Metaplastic carcinoma	1
Mucinous carcinoma	1
Normal tissue	7
Predominantly lobular	1
Spindle cell sarcoma	1
Missing	1

**Table 2 cells-12-01917-t002:** Potential oncogenic and potential tumor suppressive miRNAs with consistent diagnostic and prognostic relevance within both our patient cohort and TCGA breast cancer cohort.

miRNA in Our Dataset	Fold Change (Unpaired)	Fold Change (Paired)	miRNA in TCGA	Logrank *p*-Value	Hazard Ratio (95% Confidence Interval)	TCGA Dataset	Potential Function Category
hsa-miR-100	0.316	0.295	hsa-miR-100	0.0195	0.766 [0.613, 0.9576]	GA, *n =* 324	tumor suppressor
hsa-miR-101	0.474	0.278	hsa-miR-101-1	0.0021	0.691 [0.546, 0.8741]	HiSeq, *n =* 755	tumor suppressor
			hsa-miR-101-2	0.0278	0.805 [0.6626, 0.9769]	HiSeq, *n =* 755	
hsa-miR-193a-3p	2.070	2.244	hsa-miR-193a	0.0060	1.481 [1.1219, 1.9555]	HiSeq, *n =* 755	oncogene
hsa-miR-204	0.241	0.148	hsa-miR-204	0.0233	0.819 [0.6905, 0.9725]	HiSeq, *n =* 755	tumor suppressor
hsa-miR-205	0.246	0.180	hsa-miR-205	0.0335	0.871 [0.7667, 0.9904]	GA, *n =* 324	tumor suppressor
hsa-miR-301b	2.118	1.948	hsa-miR-301b	0.0100	1.410 [1.0837, 1.8342]	GA, *n =* 324	oncogene
hsa-miR-30a	0.406	0.324	hsa-miR-30a	0.0176	0.843 [0.7324, 0.9711]	HiSeq, *n =* 755	tumor suppressor
hsa-miR-30a*	0.351	0.250					
hsa-miR-379	0.478	0.418	hsa-miR-379	0.0480	0.797 [0.6369, 0.9981]	GA, *n =* 324	tumor suppressor
hsa-miR-615-3p	1.8673	2.531	hsa-miR-615	0.0068	1.332 [1.0804, 1.6414]	GA, *n =* 324	oncogene
hsa-miR-7	2.4657	3.994	hsa-miR-7-2	0.0393	1.464 [1.0182, 2.1057]	GA, *n =* 324	oncogene
			hsa-miR-7-3	0.0106	1.543 [1.1036, 2.1582]	GA, *n =* 324	
hsa-miR-720	4.2969	5.964	hsa-miR-720	0.0179	2.867 [1.189, 6.9117]	HiSeq, *n =* 755	oncogene
hsa-miR-99a	0.2919	0.242	hsa-miR-99a	0.0128	0.744 [0.5901, 0.939]	GA, *n =* 324	tumor suppressor

**Table 3 cells-12-01917-t003:** Pan-sensitive and pan-resistant genes and miRNAs to 19 NCCN-recommended breast cancer (BRCA) drugs with concordant expression at both mRNA and protein levels. mRNA level–italic font; protein level–normal font; both levels–bold font. Genes harboring at least one non-silent mutation in any of the BRCA cell lines were highlighted in blue font. Genes associated with fusions only in BRCA cell lines were marked in green font, while genes exhibiting both fusions and non-silent mutations in BRCA cell lines were specifically highlighted in red font. The miRNAs that were also selected in Table 2 were highlighted in yellow.

Drug	Pan-Sensitive Genes	Pan-Resistant Genes
5-fluorouracil	*GALM, HELLS, RBM14, DUOX1, RBM25, NOP56*, DHX15, BAZ1B, DGCR8, CD276, SNX13, PTPRF, PHIP, TAOK1, **SUZ12**, PLEK, hsa-miR-1246, hsa-miR-617	*KIF3A, KAT2B, MYADM, DDX3Y*, PTMS, EPHA7, EMC4, IRAK4, hsa-miR-33a, hsa-miR-1260
alpelisib	*DUOX1, PTPRF, ZBED6,* DGCR8, TRABD, NOP56, RBM25, SUZ12, **GALM**, WDR82, CDS2, hsa-miR-617, hsa-miR-933	*KAT2B, EPHA7, BCL6, MTX3,* IRGQ, TXNL1, IRAK4, GSK3A, **APOD**, **KIF3A**, MYADM, hsa-miR-1274a, hsa-miR-1280, hsa-miR-335, hsa-miR-376a, hsa-miR-720
capecitabine		*GSK3A, ZC3H4, IRGQ*, BCL6, MGST2, LZTS2
carboplatin	*WDR82*	
cisplatin	*SNX13, GATB, PRDM2, MFSD9, PHIP, TBRG1, CBX3, RPRD2, RBM14,* PTPRF, EGR1, DHX15, **PDE3A**, **ATG4D**, HERC6, DUOX1	*PTMS, PEG10*, IRGQ, **APOD**, MTX3, PTEN, STX3, **ZNF24**, **LZTS2**, KIF3A, FRYL, **MYADM**, hsa-miR-301a, hsa-miR-497, hsa-miR-335, hsa-miR-1260
cyclophosphamide	*DHX15, RBM25, PHIP*, DGCR8, PRDM2, RPRD2, PTPRF, hsa-miR-1246	*LZTS2, PTEN*, FRYL, MYADM
docetaxel	*PDE3A, TBRG1,* BAZ1B, DHX15, TBRG1, RBM25, **RPRD2**, hsa-miR-1246, hsa-miR-142-5p	*GSK3A, ZC3H4, STX3, STMN3, IRAK4, APOD, KAT2B*, EPHA7, PEG10
doxorubicin	*CDS2, HERC6, NOP56, CD276, TRABD, MFSD9*, PHIP, HELLS, GATB, **BAZ1B**, NKTR, **RBM25**, SUZ12, CBX3, CHD9, RBM14	KIF3A, PTMS, IRGQ, GSK3A, TTC26, hsa-miR-301a
epirubicin	*NKTR, RBM25, TRABD, GATB*, LSAMP, **DHX15**, PLEK	*PTEN, ZNF24*, LZTS2, EMC4, ZC3H4, MGST2, hsa-miR-1260, hsa-miR-21, hsa-miR-301a
fulvestrant	*BAZ1B, PRDM2, RBM25, CBX3, SUZ12, RBM14, HELLS, DHX15, NKTR, DGCR8*, TAOK1, GATB, TRABD, LSAMP, RPRD2, CD276, hsa-miR-1976, hsa-miR-378, hsa-miR-766, hsa-miR-1246	*MGST2, IRGQ, PSG6,* **APOD**, IRAK4, GSK3A, MYADM, hsa-miR-33a
gemcitabine	*TBRG1, NOP56, MFSD9, RBM25*, BAZ1B	*APOD*, PTMS, MAVS, ZC3H4, hsa-miR-301a
ixabepilone	*DUOX1, TBRG1, ZBED6*, RBM14, LSAMP, BAZ1B	*LZTS2, STX3, PTEN, PEG10*, KIF3A
lapatinib	*DUOX1, HERC6, PLEK*, PRDM2, CD276, hsa-miR-933	*ZNF24, TTC26, KAT2B, PTMS,* IRGQ, **TXNL1**, IRAK4, KIF3A, MYADM, PSG6
methotrexate	*MFSD9, PHIP*, RPRD2, CBX3, hsa-miR-1246	*MYADM*, LZTS2, hsa-miR-497
neratinib	*SNX13, ZBED6*, PRDM2, hsa-miR-378, hsa-miR-10b	*TXNL1, MTX3, IRGQ, TTC26, EMC4*, **ZNF24**, STMN3, hsa-miR-21
olaparib	*PTPRF, CHD9, GATB, PRDM2, BAZ1B, CBX3, PDE3A, PHIP*, ZBED6, MFSD9, **TBRG1**, CD276, SNX13	*MTX3, FRYL, BCL6*, KAT2B, LZTS2, **APOD**, DDX3Y, **MYADM**
paclitaxel	*GALM, DHX15, LSAMP*, SNX13	*MAVS, BCL6, ZC3H4, STX3, KAT2B, MTX3, KIF3A, STMN3, PEG10*, MYADM, PTMS, hsa-miR-21, hsa-miR-497
tucatinib	*CDS2, ATG4D, SNX13, PLEK*, DUOX1, GATB, CHD9, hsa-miR-205, hsa-miR-378, hsa-miR-933, hsa-miR-376c	**MTX3**, PTEN, **ZNF24**, APOD, GSK3A, hsa-miR-301a
vinorelbine	*PRDM2, DGCR8, TRABD, EGR1, BAZ1B, CD276, PTPRF, GATB, ATG4D, TBRG1, TAOK1, WDR82, SUZ12, RBM14*, NKTR, PDE3A, **RPRD2**	MTX3, MAVS, TTC26, EPHA7, hsa-miR-21, hsa-miR-199b-5p

**Table 4 cells-12-01917-t004:** Sensitive and resistant genes with concordant mRNA and protein expression associated with response to the new/repositioning drugs identified with CMap. Genes harboring at least one non-silent mutation in any of the BRCA cell lines were highlighted in blue font. Genes associated with fusions only in BRCA cell lines were marked in green font, while genes exhibiting both fusions and non-silent mutations in BRCA cell lines were specifically highlighted in red font.

Drug	Sensitive Gene	Resistant Gene
Lestaurtinib	FHL1, FKBP7, LAMA4, LAMB3, LAMC1, MMP14	ARRB1
BRD-K12244279	ACSL4, ASPH, CAV1, CAV2, EHD2, FMNL2, FSTL1, FYN, IFIT1, IFIT2, LDHB, SAMD9, UPP1, WLS	ENPP1
PD198306	ICAM1, IFIT3	
Pilocarpine	SDC2	
TG101348	VLDLR	ANXA8, APOL1, FBLN1, ICAM1, MX2
Tremorine	CRIP1, GPC4	
U0126	DKK3, FSD1, KYNU, LRP12	
Y-27632	C1R, CASP1, FOSL1, GSDMD	CLIC3, CPLX1, SYAP1

## Data Availability

All data are publicly available with accession information provided in the manuscript and Appendix A.

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
