# Peer review of "MicroRNA-Based Discovery of Biomarkers, Therapeutic Targets, and Repositioning Drugs for Breast Cancer"

_cells, 2023, doi:10.3390/cells12141917_

Round 1

Reviewer 1 Report

In their review paper the authors summarize the available public ressources for drug sensitivity, RNA/miRNA data and proliferation data of breast cancer from patients as well as from breast cancer tumor cell lines. They analyzed theses data in silico identifying potential oncogenic and tumor suppressive miRNAs associated with prognosis and resistance or miRNAs differentiating normal and cancerous tissue.

The weekness of this study is the low number of tumor samples in different subgroups. According to table 1 there is only one case with DCIS, but Figure 4 and 5 compare DCIS with normal and with invasive breast cancer. Statistics with one case is not possible. Which cases are included when giving invasive breast cancer? They have to declare which breast cancer subtypes are invasive? HER2Neu ++ or TNBC? Etc. Finally the Figure 1 showing the heatmap is confusing.

The paper is not focussed, there are many different comparisons, but none of them is really given in detail. A better publication strategy could be the focus on resistance only. Here is potential for novelty. Table 3 includes very interesting data. The miRNAs that are listed in pan-sensitive group, do they have targets in the pan-resistant genes group and vice versa? This association needs to be specified. Red and orange is difficult to recognize. Figure 8 needs better explanation. What they discovered from this analysis? They should link in a graph the Table 4 with targets found in Table 3.

The authors should give which genes are connected with which pathways (lines 343-348)?

The supplementary Tables but also Tables 3 and 4 include a lot of primary data, but for a better understanding of the paper there is necessary to give more and valid analysis results and to extract novelty. The reader is not willing and may be able to extract the possible interesting information from such original data tables by himself.

In general the data should be presented in a more attractive and skillful way.

Minor:

The authors want to show common miRNAs between TCGA and Violina et al. Study, why not use a VENN diagram to show the overlapping miRNA set more reader friendly?

na

Author Response

Reviewer 1

Comments and Suggestions for Authors

Reviewer: In their review paper the authors summarize the available public resources for drug sensitivity, RNA/miRNA data and proliferation data of breast cancer from patients as well as from breast cancer tumor cell lines. They analyzed theses data in silico identifying potential oncogenic and tumor suppressive miRNAs associated with prognosis and resistance or miRNAs differentiating normal and cancerous tissue.

Authors: We appreciate the thorough critique from the reviewer. The constructive comments are very helpful to improve our manuscript.

Reviewer: The weakness of this study is the low number of tumor samples in different subgroups. According to table 1 there is only one case with DCIS, but Figure 4 and 5 compare DCIS with normal and with invasive breast cancer. Statistics with one case is not possible. Which cases are included when giving invasive breast cancer? They have to declare which breast cancer subtypes are invasive? HER2Neu ++ or TNBC? Etc. Finally the Figure 1 showing the heatmap is confusing.

Authors: We agree with these comments. Previous Figures 4 and 5 are removed from the manuscript due to the single sample in the DCIS group. In Figure 1, four breast cancer samples were mistakenly classified into the normal breast tissue group, and one normal breast tissue sample was misclassified into the breast cancer group. The overall accuracy in the unsupervised classification was 90.4%. This is now added in section 3.1.  

Reviewer: The paper is not focused, there are many different comparisons, but none of them is really given in detail. A better publication strategy could be the focus on resistance only. Here is potential for novelty. Table 3 includes very interesting data. The miRNAs that are listed in pan-sensitive group, do they have targets in the pan-resistant genes group and vice versa? This association needs to be specified. Red and orange is difficult to recognize. Figure 8 needs better explanation. What they discovered from this analysis? They should link in a graph the Table 4 with targets found in Table 3.

Authors: To be more focused, previous Figures 6 and 7 are replaced with a Venn Diagram (Figure 4) and details are included in Supplementary File 1. A new Figure 5 is now added to show the pan-sensitive miRNA-targeted genes that are pan-resistant, and vice versa (all listed in Table 3). The orange color in Table 3 is now changed to green. The pipeline for compound screening is now added to Figure 8 (now Figure 6). This analysis discovered drugs with the potential to improve breast cancer treatment outcomes. From pan-sensitive and pan-resistant genes in Table 3, together with proliferation genes and survival hazard/protective genes, a 56-gene expression signature was constructed as CMap input to discover new drugs for treating breast cancer. Patient responders/non-responders for these new drug options were characterized in Table 4. This linkage is now added in the new Figure 5.   

Reviewer: The authors should give which genes are connected with which pathways (lines 343-348)?

Authors: Genes for each pathway are added in the manuscript and Supplementary File 3.

Reviewer: The supplementary Tables but also Tables 3 and 4 include a lot of primary data, but for a better understanding of the paper there is necessary to give more and valid analysis results and to extract novelty. The reader is not willing and may be able to extract the possible interesting information from such original data tables by himself.

Authors: Significantly enriched cytobands and gene families for pan-sensitive and pan-resistant genes in Table 3 were identified with ToppGene. No significant pathways were found. These results are now added to the manuscript and Supplementary File 2.

After adding patient survival and proliferation genes to the analysis, we got a refined list of 56 genes. Through the CMap analysis, significant pathways and compound sets were discovered and described in the manuscript.

Reviewer: In general the data should be presented in a more attractive and skillful way.

Authors: This comment has been addressed.

Minor:

Reviewer: The authors want to show common miRNAs between TCGA and Violina et al. Study, why not use a VENN diagram to show the overlapping miRNA set more reader friendly?

Authors: A Venn Diagram (Figure 4) is now added to the manuscript, showing overlapping genes among our cohort, Iorio et al (Violina et al was found to be a wrong citation and is now corrected), and TCGA. Supplementary File 1 is also updated to include the detailed association between gene expression and clinical phenotypes. The previous figures are now removed.

The results in Iorio et al were revised after removing 14 breast cancer cell lines from the analysis of breast cancer tumors vs. normal breast tissues, leaving only 76 neoplastic breast tissues in the comparison with 7 normal breast tissue samples.

Reviewer 2 Report

Introduction:

Introduction is clearly providing relevant information on breast cancer, highlighting the need for improved early detection method, and personalised treatments. However, there are some mistakes of coherence, grammar and spelling which should be taken care of.

Methodology:

1-In methodology section, all the protocols have been defined well however they could be better explained specifically as well for specific implementation.

In microarray section, While the article mentions that miRNA profiling was conducted using custom microarrays, it does not provide information about the specific platform used or the technical specifications of the microarrays. Additionally, there is no mention of the normalization method employed or the criteria used for data processing and analysis. Providing these details would improve the transparency of the microarray analysis.

2-The article mentions the use of several bioinformatic tools, such as TarBase database and Connectivity Map (CMap), but does not provide sufficient details about how these tools were used or the specific criteria employed for data interpretation. Providing more information about the parameters used and the rationale behind tool selection would enhance the credibility of the bioinformatics analysis.

By addressing these shortcomings and providing more specific and detailed information, the methodology section of the article would become more robust, transparent, and reproducible, strengthening the overall scientific rigor of the study.

Results:

 1-The article does not provide information about the size of the patient cohort used in the study. A small sample size may limit the generalizability of the findings and may not adequately represent the broader population of breast cancer patients.

2-Although the article mentions validation in an external patient cohort, it does not provide detailed information about the size or characteristics of the validation cohort. Without robust external validation, the reliability and reproducibility of the findings are uncertain.

 3-The article mentions fold changes of miRNAs in tumor tissues, normal tissues, and blood samples. However, it does not provide detailed information on the magnitude of these fold changes or their clinical significance. Without this information, it is challenging to interpret the biological relevance of the observed differential expression.

 4-The article mentions the discovery of potential new drug options through the identification of target genes associated with drug response. However, it lacks a thorough discussion of the criteria and rationale used for selecting these specific drugs. Providing more details about the drug repositioning process and its potential limitations would strengthen the article's findings.

 5-The article could benefit from a more comprehensive discussion of the limitations and potential challenges associated with miRNA-based biomarkers and drug repositioning. Additionally, suggestions for future research directions and the integration of other -omics data, such as genomics or proteomics, could enhance the article's impact and encourage further scientific exploration.

Overall, while the study identifies differentially expressed miRNAs and provides initial evidence of their potential diagnostic and prognostic value in breast cancer, further validation and detailed analysis are needed to establish their clinical utility and understand their functional significance.

 English language throughout the manuscript needs to be checked to correct spelling, punctuation and grammar errors.

English language throughout the manuscript needs to be checked to correct spelling, punctuation and grammar errors. 

Author Response

Reviewer 2

Comments and Suggestions for Authors

Introduction:

Reviewer: Introduction is clearly providing relevant information on breast cancer, highlighting the need for improved early detection method, and personalised treatments. However, there are some mistakes of coherence, grammar and spelling which should be taken care of.

Authors: We appreciate the positive review and constructive comments. The grammatical mistakes are now corrected.

Methodology:

Reviewer: 1-In methodology section, all the protocols have been defined well however they could be better explained specifically as well for specific implementation.

In microarray section, While the article mentions that miRNA profiling was conducted using custom microarrays, it does not provide information about the specific platform used or the technical specifications of the microarrays. Additionally, there is no mention of the normalization method employed or the criteria used for data processing and analysis. Providing these details would improve the transparency of the microarray analysis.

Authors: The custom miRNA microarray platform is Sanger 15. In normalization, a detection threshold was determined for each array by calculating the sum of five times the standard deviation of the background signal and the 10% trim mean of the negative control probes. Probes exhibiting consistently low signals were eliminated from the subsequent statistical analysis using these threshold values. The raw and normalized miRNA data and the miRNA platform details are available in the NCBI GEO with the accession number GSE37963. The details are now added in section 2.3.

Reviewer: 2-The article mentions the use of several bioinformatic tools, such as TarBase database and Connectivity Map (CMap), but does not provide sufficient details about how these tools were used or the specific criteria employed for data interpretation. Providing more information about the parameters used and the rationale behind tool selection would enhance the credibility of the bioinformatics analysis.

By addressing these shortcomings and providing more specific and detailed information, the methodology section of the article would become more robust, transparent, and reproducible, strengthening the overall scientific rigor of the study.

Authors: The bioinformatics pipeline description is now added. Detailed parameters used in the result selection are now provided in the manuscript.  

Results:

Reviewer: 1-The article does not provide information about the size of the patient cohort used in the study. A small sample size may limit the generalizability of the findings and may not adequately represent the broader population of breast cancer patients.

Authors: The sample size for each patient cohort is now specified where applicable. We acknowledge the small sample size as a limitation of this study.

Reviewer: 2-Although the article mentions validation in an external patient cohort, it does not provide detailed information about the size or characteristics of the validation cohort. Without robust external validation, the reliability and reproducibility of the findings are uncertain.

Authors: A description of the external patient cohort is now added in section 2.1. We corrected the citation of this external patient cohort. The results in the external cohort from Iorio et al were revised after removing 14 breast cancer cell lines from the analysis of breast cancer tumors vs. normal breast tissues, leaving only 76 neoplastic breast tissues in the comparison with 7 normal breast tissue samples.

Reviewer: 3-The article mentions fold changes of miRNAs in tumor tissues, normal tissues, and blood samples. However, it does not provide detailed information on the magnitude of these fold changes or their clinical significance. Without this information, it is challenging to interpret the biological relevance of the observed differential expression.

Authors: We thank the reviewer for the comment. The classification accuracy in these comparisons is now added to the results section where applicable. More detailed fold change information is now provided in Supplementary Files. 

Reviewer: 4-The article mentions the discovery of potential new drug options through the identification of target genes associated with drug response. However, it lacks a thorough discussion of the criteria and rationale used for selecting these specific drugs. Providing more details about the drug repositioning process and its potential limitations would strengthen the article's findings.

Authors: To identify candidate compounds for treating breast cancer, we designed the mechanism of drug action to maintain the expression of non-proliferative survival-protective/pan-sensitive genes and suppress the survival-hazard/pan-resistant genes. Among the 4,117 genes targeted by our diagnostic miRNAs, we performed a refined selection to identify 28 up-regulated and 28 down-regulated gene lists for input into the CMap analysis. The 28 up-regulated genes met the following criteria: 1) exhibited a significant hazard ratio < 1 (log-rank p < 0.05) in survival analysis of TCGA-BRCA; 2) were classified as pan-sensitive genes for all 19 NCCN breast cancer drugs at both the mRNA and protein levels in CCLE BRCA cell lines; 3) did not display significant dependency scores (< –0.5) in more than 50% of the BRCA cell lines in both CRISPR-Cas9 and RNAi screening data. The 28 down-regulated genes met the following criteria: 1) demonstrated a significant hazard ratio > 1 (log-rank p < 0.05) in survival analysis of TCGA-BRCA; 2) were classified as pan-resistant genes for all 19 NCCN breast cancer drugs at both the mRNA and protein levels in CCLE BRCA cell lines.

We input the selected 28 up-regulated genes and the 28 down-regulated genes into CMap. From the CMap output, we selected significant (raw connectivity score > 0.9, p < 0.05) pathways and compound sets as heuristic therapeutic targets. We further examined the half-maximal inhibitory concentration (IC50) and half-maximal effective concentration (EC50) of the significant compounds in the PRISM dataset. Our pipeline for repositioning drug discovery is now shown in Figure 7A.

Through the CMap analysis, compounds that can inhibit the input down-regulated genes and maintain the expression of the input up-regulated genes in breast cancer cells were identified. The significant pathways and compound sets (raw connectivity score > 0.9, p < 0.05) were considered valid hypotheses worthy of further investigation. From these significant results, we further examined the compounds with average IC50/EC50 measurements less than 7 µM in human breast cancer cell lines as potential drugs for breast cancer treatment.  In this study, we selected 20 potential new or repurposed drugs with available IC50/EC50 measurements in the PRISM dataset.   

These descriptions are now added in the results and discussion sections.

Reviewer: 5-The article could benefit from a more comprehensive discussion of the limitations and potential challenges associated with miRNA-based biomarkers and drug repositioning. Additionally, suggestions for future research directions and the integration of other -omics data, such as genomics or proteomics, could enhance the article's impact and encourage further scientific exploration.

Authors:

The AI pipeline presented in this study can also be applied to model other data types for new/repositioning drug discovery, such as DNA copy number variation, transcriptomic, and proteomics profiles in bulk tumors and single cells for other cancer types as we previously published [1-4]. In the future, we will also apply this pipeline to model other structural variants for the discovery of biomarkers and therapeutic targets.

This study has several limitations. First, our patient cohort has a small sample size. The identified tissue-based 86 miRNAs and six blood-based miRNAs can separate breast cancer and normal samples with an overall accuracy of 90.4% and 100%, respectively. Nevertheless, due to the small sample size, their clinical utility in breast cancer diagnosis needs to be substantiated with the following studies: (1) an independent external patient cohort with a larger sample size; (2) high overall accuracy, specificity, and sensitivity in the external validation with the classification model fixed in the training-validation process; and (3) prospective validation of clinical benefits showing improved patient outcomes by using the diagnostic miRNA tests. Fulfilling these tasks will be time-consuming and requires a vast amount of resources. Second, the functions of the identified potential oncomiRs and potential tumor-suppressing miRNAs need to be investigated in future research. Following the mechanistic characterization, the development of miRNA-based drugs needs to overcome technical challenges in achieving the optimal delivery route, intracellular efficacy, tissue specificity, and in-vivo stability. Finally, the identified compounds as new or repositioning drugs for breast cancer treatment were based on bioinformatics analysis and in-vitro IC50/EC50 measurements in human breast cancer cell lines. Their efficacy in treating breast cancer needs to be tested in future animal studies and/or clinical trials before obtaining regulatory approvals for clinical applications.  

This is added to the Discussion.

Reviewer: Overall, while the study identifies differentially expressed miRNAs and provides initial evidence of their potential diagnostic and prognostic value in breast cancer, further validation and detailed analysis are needed to establish their clinical utility and understand their functional significance.

Authors: We agree. The discussion is provided in the above section and is added to the manuscript.

Reviewer: English language throughout the manuscript needs to be checked to correct spelling, punctuation and grammar errors.

Authors: Corrected.

References

  1. Ye, Q.; Falatovich, B.; Singh, S.; Ivanov, A.V.; Eubank, T.D.; Guo, N.L. A Multi-Omics Network of a Seven-Gene Prognostic Signature for Non-Small Cell Lung Cancer. International journal of molecular sciences 2021, 23, doi:10.3390/ijms23010219.
  2. Ye, Q.; Guo, N.L. Single B Cell Gene Co-Expression Networks Implicated in Prognosis, Proliferation, and Therapeutic Responses in Non-Small Cell Lung Cancer Bulk Tumors. Cancers 2022, 14, 3123.
  3. Ye, Q.; Hickey, J.; Summers, K.; Falatovich, B.; Gencheva, M.; Eubank, T.D.; Ivanov, A.V.; Guo, N.L. Multi-Omics Immune Interaction Networks in Lung Cancer Tumorigenesis, Proliferation, and Survival. International journal of molecular sciences 2022, 23, 14978.
  4. Ye, Q.; Singh, S.; Qian, P.R.; Guo, N.L. Immune-Omics Networks of CD27, PD1, and PDL1 in Non-Small Cell Lung Cancer. Cancers 2021, 13, 4296, doi:10.3390/cancers13174296.

Round 2

Reviewer 1 Report

the authors answered to my questions, but not always in a satisfying way. The manuscript

quality has been improved but manuscript is still not ready for publishing.

Reviewer: The weakness ofthis study is the low number oftumor samples in different

subgroups. According to table 1 there is only one case with DCIS, but Figure 4 and 5 compare

DCIS with normal and with invasive breast cancer. Statistics with one case is not possible.

Which cases are included when giving invasive breast cancer? They have to declare which

breast cancer subtypes are invasive? HER2Neu ++ or TNBC? Etc. Finally the Figure 1

showing the heatmap is confusing.

Authors: We agree with these comments. Previous Figures 4 and 5 are removed from the

manuscript due to the single sample in the DCIS group. In Figure 1, four breast cancer

samples were mistakenly classified into the normal breast tissue group, and one normal breast

tissue sample was misclassified into the breast cancer group. The overall accuracy in the

unsupervised classification was 90.4%. This is now added in section 3.1.

-they did not answer regarding the invasive carcinoma subtypes,

Figure 4 is the Venn diagram, it is appreciated to show such data, miRNA 204 has been

selected as the most important one, what is the role of miRNA 204, what is known about this

miRNA 204? in discussion is only one sentence. Does this miRNA play a role in drug

resistance also? what are the major targets for this miRNA?

The first and second part of the manuscript are still not linked. They should highlight in Table

3 miRNAs or targets that have been identified by comparison between tumor and non-tumor

samples.

Author contribution: give, who has performed the revision

Author Response

Comments and Suggestions for Authors

Reviewer: the authors answered to my questions, but not always in a satisfying way. The manuscript

quality has been improved but manuscript is still not ready for publishing.

Reviewer: The weakness of this study is the low number of tumor samples in different

subgroups. According to table 1 there is only one case with DCIS, but Figure 4 and 5 compare

DCIS with normal and with invasive breast cancer. Statistics with one case is not possible.

Which cases are included when giving invasive breast cancer? They have to declare which

breast cancer subtypes are invasive? HER2Neu ++ or TNBC? Etc. Finally the Figure 1

showing the heatmap is confusing.

Authors: We agree with these comments. Previous Figures 4 and 5 are removed from the

manuscript due to the single sample in the DCIS group. In Figure 1, four breast cancer

samples were mistakenly classified into the normal breast tissue group, and one normal breast

tissue sample was misclassified into the breast cancer group. The overall accuracy in the

unsupervised classification was 90.4%. This is now added in section 3.1.

Reviewer: -they did not answer regarding the invasive carcinoma subtypes,

Authors: We removed all the analyses that referred to invasive breast carcinoma. The denoted breast cancer in Figure 1 referred to cases with histology of ductal adenocarcinoma, invasive ductal carcinoma, invasive lobular carcinoma, metaplastic carcinoma, mucinous carcinoma, and predominantly lobular in Table 1. This description is now added in results section 3.1.

Reviewer: Figure 4 is the Venn diagram, it is appreciated to show such data, miRNA 204 has been

selected as the most important one, what is the role of miRNA 204, what is known about this

miRNA 204? in discussion is only one sentence. Does this miRNA play a role in drug

resistance also? what are the major targets for this miRNA?

Authors: MiR-204 was identified as a potential tumor suppressor as under-expressed in tumors in our patient cohort and samples from Iorio et al. as well as patients with a poor prognosis in TCGA-BRCA (Table 2 and Figure 4). MiR-204 was associated with drug sensitivity to carboplatin and tucatinib, and drug resistance to docetaxel, ixabepilone, paclitaxel, and vinorelbine. In addition, miR-204 was associated with resistance to RITA, a potential new drug option for treating breast cancer (Supplementary File S2). MiR-204-3p and miR-204-5p targeted genes were provided in Supplementary File S2. Among these target genes, those that were pan-sensitive or pan-resistant to 19 NCCN-recommended breast cancer drugs (Table 3) were shown in Figure 8. MiR-204 targeted genes that were sensitive to  BRD-K12244279 (Table 4) were also shown in Figure 8.

These results are now added to the Results section 3.3 and Supplementary File S2.

Reviewer: The first and second part of the manuscript are still not linked. They should highlight in Table

3 miRNAs or targets that have been identified by comparison between tumor and non-tumor

samples.

Authors: MiR-205 identified as a potential tumor suppressor in Table 2, was also found as pan-sensitive to 19 NCCN-recommended breast cancer drugs in Table 3 (now highlighted). MiR-301b, an identified potential oncomiR in Table 1, has a family member miR-301a as pan-resistant to 19 NCCN-recommended drugs. This description is now added to the Discussion.

Reviewer: Author contribution: give, who has performed the revision

Authors: Drs. Qing Ye and Nancy Lan Guo performed the revision. It is now added in the Author contribution.

Reviewer 2 Report

The author's have been answered all questions as requested by the reviewers. However, the reference number 33 should be changed and corrected.  

Author Response

We thank the reviewer for raising this issue. This reference is fixed.

Round 3

Reviewer 1 Report

The authors answered to my questions in a sufficient way. The manuscript is improved. Part 1 and part 2 are now better linked, novelty of data is now more clear to the reader.